# Prevalence of Tuberculosis in Central Asia and Southern Caucasus: A Systematic Literature Review [note 1]

**DOI:** 10.3390/diagnostics15182314

**Published:** 2025-09-12

**Authors:** Malika Idayat, Elena von der Lippe, Nailya Kozhekenova, Oyunzul Amartsengel, Kamila Akhmetova, Ainash Oshibayeva, Zhansaya Nurgaliyeva, Natalya Glushkova

**Affiliations:** 1The Department of Epidemiology, Biostatistics and Evidence-Based Medicine, Faculty of Medicine and Health Care, Al-Farabi Kazakh National University, Almaty 050040, Kazakhstan; nailyakozhekenova@gmail.com (N.K.); zhanlazim@gmail.com (Z.N.); glushkovanatalyae@gmail.com (N.G.); 2The Faculty of Postgraduate Medical Education, Khoja Akhmet Yassawi International Kazakh-Turkish University, Turkistan 161200, Kazakhstan; ainash.oshibayeva@ayu.edu.kz; 3Department of Epidemiology and Health Monitoring, Robert Koch Institute, 13353 Berlin, Germany; vonderlippee@rki.de; 4Department of Health Policy, Mongolian National University of Medical Science, Ulaanbaatar 14210, Mongolia; oyunzul.md@gmail.com; 5Department of Public Health and Management, NJSC «Astana Medical University», Astana 010000, Kazakhstan; akhmetova.km@amu.kz

**Keywords:** tuberculosis, prevalence, Central Asia, Southern Caucasus, Mongolia

## Abstract

**Background**: In 2023, tuberculosis (TB) caused 1.25 million deaths globally, remaining a leading infectious killer. Central Asia and Southern Caucasus face high TB burdens, particularly Mongolia. This review synthesizes TB prevalence data and diagnostic capabilities in these regions to support public health strategies. **Methods**: This systematic review aimed to synthesize current data on TB prevalence in Central Asia, Southern Caucasus, and Mongolia to support public health strategies and research priorities. A comprehensive search of PubMed and Google Scholar was conducted for English-language articles published up to 2023. Studies were assessed using a modified Newcastle–Ottawa Scale. Nine studies met the inclusion criteria, covering Kazakhstan, Kyrgyzstan, Uzbekistan, Tajikistan, Turkmenistan, Mongolia, Georgia, Armenia, and Azerbaijan. **Results**: TB incidence ranged from 67 per 100,000 in Kazakhstan to 190 per 100,000 in Kyrgyzstan, with the highest prevalence of 68.5% in Mongolia. TB affected men more frequently (65.3%), and the key risk factors included HIV (30.5%), comorbidities, and undernutrition. Diagnostic performance varied significantly (microscopy sensitivity, 45–65%; GeneXpert MTB/RIF, 89–96% sensitivity and 98% specificity for rifampicin resistance). Diagnostic turnaround times ranged from hours (molecular) to weeks (conventional). Only 58% of TB facilities had GeneXpert technology, with urban–rural disparities in diagnostic access. Drug-resistant TB imposed a significant economic burden, with treatment costs ranging from USD 106 to USD 3125. **Conclusions**: Strengthening surveillance, improving data collection, and conducting longitudinal studies are essential for designing effective TB control strategies in these regions. Significant diagnostic gaps persist across these regions, especially with regard to drug-resistant strains. Point-of-care molecular diagnostics, improved algorithms, and expanded laboratory training show promise. Future research should focus on rapid biomarker-based diagnostics, field-deployable technologies for settings with limited resources, and AI integration to enhance diagnostic accuracy and efficiency.

## 1. Introduction

Tuberculosis (TB) is a major contributor to the global burden of infectious diseases, causing over one million deaths annually. The World Health Organization (WHO) European Region accounts for 3% of global TB cases, with significant variations in disease burden across countries [1,2]. Controlling the spread of TB remains a top priority for global health efforts. The concept of the “burden of tuberculosis” is used to describe the epidemiological situation of the disease, as it combines morbidity (incidence and prevalence) and mortality data into a single comprehensive indicator [3,4].

According to the WHO, 10.8 million new cases of TB were reported in 2023, resulting in 1.25 million deaths [5]. This factor accounts for 1.86% of the total global disability-adjusted life years (DALYs), resulting in a loss of 2.54% of total years of life lost (YLL). Consequently, TB ranks 12th and 11th in terms of global health importance in these respective categories [6].

The TB epidemic varies globally, affecting all countries and age groups, though 90% of cases occur in adults, with 9% among people with HIV (72% in Africa). Two-thirds of cases are concentrated in India (27%), China (9%), Indonesia (8%), the Philippines (6%), Pakistan (5%), Nigeria (4%), Bangladesh (4%), and South Africa (3%), with 87% of all cases found in these and 22 other WHO high-burden countries. Only 6% of cases are in the WHO European and Americas regions. Developed countries show a decline; in the USA, TB incidence dropped from 9.7 to 2.2 per 100,000 from 1993 to 2020. In South Korea, the economic burden of multidrug-resistant tuberculosis (MDR-TB) varies, with higher DALY rates among middle-aged individuals. In 2017, TB accounted for 6.5% of the total DALY index, rising to 17% among those over 65 [7,8].

TB remains a major public health concern in Central Asia. Collectively, Kazakhstan, Kyrgyzstan, Tajikistan, Turkmenistan, and Uzbekistan report over 34,000 TB cases and 8000 drug-resistant TB (DR-TB) cases annually [9]. According to estimates, in 2023, the highest TB incidence rate in the WHO European Region was in Kyrgyzstan and Ukraine (112 per 100,000 population), followed by Tajikistan (79 cases per 100,000 population), the Republic of Moldova (76 cases per 100,000 population), and Kazakhstan (70 cases per 100,000 population) [10]. Armenia, Azerbaijan, and Georgia, the three countries comprising the Southern Caucasus, are among the 18 high-priority countries (HPCs) in Eastern Europe and Central Asia that account for 85% of TB incidence and over 90% of drug-resistant TB cases in the WHO European Region [11].

Although there have been substantial decreases in case notifications over the past decade, TB remains a significant public health issue in Southern Caucasus. During the period 2014–2023, the HPCs with the highest annual rate of decline were Latvia (−10.4%), Estonia (−9.9%), Armenia (−8.8%), Lithuania (−7.8%), Bulgaria (−7.4%), Georgia (7.0%), and Belarus (6.8%) [10]. The incidence of TB in 2023 in Armenia equaled 25 per 100,000 population, in Azerbaijan 72 per 100,000 population, and Georgia notified 55 per 100,000 population. The rate in most EU countries is under 10 per 100,000 population [12].

The TB death rate in Mongolia has remained stable at 10.0 cases per 100,000 people over the last 3 years [13]. Mongolia, a lower-middle-income country of 3 million people, has the fourth-highest TB incidence rate in the WHO Western Pacific Region, at 428 cases per 100 000 people, and one of the world’s lowest TB treatment coverage rates, at 31% [14].

Globally, key risk factors for TB include biological, socioeconomic, and behavioral aspects. HIV co-infection, diabetes, and previous TB treatment significantly increase susceptibility, while poverty, overcrowding, smoking, and alcohol consumption further elevate the risk [15]. Several studies have examined the influence of risk factors on TB prevalence in different groups. For instance, a study by Daniel E. et al. in Tajikistan identified 59 cases of active pulmonary TB among prisoners, with a point prevalence of 4.5% (95% CI: 3.4–5.7) [16]. Similarly, Tilloeva Z. et al. investigated TB among a key population group, reporting that 29.8% of patients had a positive sputum smear, 14.1% were drug-sensitive, and 11.3% had mono-DR/MDR-TB [17].

The COVID-19 pandemic has had a significant negative impact on efforts to combat TB [18]. Reduced diagnosis and treatment, due to restrictions and overburdened health systems, have led to increased mortality and more severe cases [19,20]. In regions with high TB incidence, the situation has further deteriorated due to the economic repercussions of the pandemic [21]. Additionally, patients co-infected with COVID-19 and TB face an increased risk of death. Logistical challenges, such as rising transport costs, have also complicated the implementation of TB control programs [20]. During the COVID-19 pandemic, there was a sharp increase in air and sea transportation costs, leading to higher import expenses, which may negatively impact TB control programs [22].

Despite increased costs and a decline in the global incidence of TB from 2000 to 2017, many TB control programs continue to experience funding shortages, particularly in low- and lower-middle-income countries [23,24]. Uzbekistan is an example of this financial strain, where, between 2016 and 2020, the annual cost of purchasing medicines for the recommended 6-month treatment course for HPV-TB and the 20-month course for MDR-TB using oral therapy averaged USD 1.4 million (±USD 274,000), with an additional USD 34,000 (±USD 6400) per year allocated for the import of medicines [22]. A similar challenge exists in Kazakhstan, where the estimated cost of diagnosing and treating a TB patient for the current episode is USD 929, while, for MDR-TB patients, it rises to USD 3125. These financial burdens highlight how the cost of anti-tuberculosis drug regimens can directly impact the number of individuals receiving treatment, potentially limiting access to life-saving therapies.

It is important to note that the situation regarding TB in Central Asia remains a significant problem. Although the absolute number of cases has decreased, the prevalence of TB in the region is still extremely high compared with Europe and America, presenting substantial challenges for both governments and patients. This is partly attributed to the socioeconomic conditions of the population and the lack of medical facilities [25]. Although some individual studies and global reports on the prevalence of TB have been conducted, there has been no comprehensive systematic review addressing the broader range of effects of TB in Central Asia [25].

Despite a decline in TB incidence in many parts of the world, the burden of TB remains a significant public health challenge in Central Asia, the Southern Caucasus, and Mongolia. Most of the countries in this study, except Mongolia, were part of the former Soviet Union and share a common healthcare legacy, including centralized TB control programs, diagnostic approaches, and systemic healthcare challenges that persist today. While Mongolia is not a former Soviet republic, it had strong political and economic ties with the USSR and adopted a similar healthcare model, making it a relevant case for comparison.

This study focuses on these specific countries because they continue to face high TB prevalence, drug-resistant TB strains, and socioeconomic factors that hinder effective TB control [26]. Unlike other post-Soviet states, such as the Baltic countries or Eastern European nations, which have significantly lower TB rates and stronger healthcare infrastructures, the selected regions remain among the most affected. Therefore, the purpose of this systematic review is to analyze the prevalence, risk factors, and healthcare challenges associated with TB in Central Asia, the Southern Caucasus, and Mongolia, providing a comprehensive regional perspective and country comparison of the ongoing TB epidemic.

## 2. Materials and Methods

### 2.1. Study Design

The analysis was conducted based on a systematic review of data regarding the prevalence of TB in Central Asia, Southern Caucasus, and Mongolia. The study was reported in accordance with the PRISMA checklist, which is presented in Appendix A.

### 2.2. Protocol and Registration

The research protocol was registered on the PROSPERO website (https://www.crd.york.ac.uk/PROSPERO/view/CRD420251020764). The registration number is PROSPERO 2025 CRD420251020764.

### 2.3. Eligibility Criteria

The results of a study reflecting the prevalence of TB in the Central Asian countries of Kyrgyzstan, Uzbekistan, Tajikistan, Kazakhstan, and Turkmenistan, in Mongolia, and in the Southern Caucasus countries of Armenia, Azerbaijan, and Georgia are presented. The study population included patients diagnosed with TB. Given the high incidence of TB in the region, studies reporting the prevalence of multidrug-resistant TB (MDR-TB) among TB patients were also accepted.

The language of the studies was either English or Russian, as international studies are primarily published in English, while national journals in Central Asia and Southern Caucasus are published in Russian.

The review included full-text scientific articles (excluding conference abstracts) on the prevalence of TB (model, population, or survey) published in peer-reviewed journals or national reports and reviews available from official government sources. Despite multiple attempts through academic databases and interlibrary services, some full-text articles were not accessible via institutional subscriptions or in open-access format. All articles published from the beginning of the database search, spanning from 2013 to 2023, were included.

### 2.4. Sources of Information

Databases including PubMed and Google Scholar were used for the search. We limited the search to articles published within the last 11 years, from 2013 to 2023 (Appendix A).

### 2.5. Search

The following keyword combinations were used in the database search: “prevalence”, “burden”, “economy”, “Tuberculosis”, “Multidrug resistance”, “Pulmonary tuberculosis”, “Mycobacterium tuberculosis”, “Central Asia”, “Southern Caucasus”, “Kazakhstan”, “Tajikistan”, “Kyrgyzstan”, “Uzbekistan”, “Turkmenistan” “Mongolia”, “Armenia”, “Azerbaijan”, and “Georgia”.

### 2.6. Selecting Articles

The researchers reviewed the titles and abstracts of all retrieved articles to assess whether they met the predefined inclusion or exclusion criteria for the study. The full texts of articles that met these criteria, based on the initial review of titles and abstracts, were then carefully examined, and reassessed to determine their final inclusion in the study, in line with the established criteria. The entire process of selecting articles for the study was documented using the PRISMA flow chart (Figure 1).

### 2.7. Data Elements

The following information was extracted from each eligible article or report: first author details, year of publication, study design, sample size, setting (country), population characteristics (including age and sex), disease characteristics (including type of TB), and disease prevalence. Prevalence data were extracted and are reported separately.

### 2.8. Quality Assessment

We assessed the quality of the articles using the Newcastle–Ottawa Scale (NOS) for cross-sectional studies, as outlined in Table 1. This scale ranges from 0 to 10 points and includes criteria for evaluating selection (up to 5 points), comparability (up to 2 points), and presentation of results (up to 3 points).

#### Quality of Studies

The quality assessment using the NOS is presented in Table 2. The overall quality was deemed satisfactory, with a mean score of 5.67 (standard deviation (SD) > 1.46) out of 10 for all studies combined. In the screening section, the mean score was 2.33 (SD ≈ 0.50) out of 5, with only two studies [27,28] achieving the maximum score of 8. In the comparability section, the mean score was 1 (SD ≈ 0.50) out of a possible 2. In the results section, the mean score was 1.89 (SD = 0.78) out of 3. After the quality assessment, 9 articles with a score higher than 4 were included in this review.

### 2.9. Summary Measures

The primary outcome measure was TB prevalence. Data were stratified by demographic characteristics, including age, sex, country, and risk factors, as well as healthcare costs, and are presented in a narrative format.

## 3. Results

### 3.1. Study Selection

As a result of the search and selection process (PRISMA, Figure 1), 131 publications remained after duplicates were removed and were considered for analysis. Following the application of inclusion criteria, 47 articles were selected, of which 13 were included in the final analysis after applying the exclusion criteria. After assessing the quality of the studies using the NOS, nine articles were selected for detailed analysis.

### 3.2. Prevalence of TB in Central Asia

In Central Asia and Southern Caucasus, studies have been conducted to assess the prevalence of TB, considering demographic characteristics and risk factors. Research on drug-resistant forms of TB has been carried out in several countries in this region, including Kazakhstan, Uzbekistan, Kyrgyzstan, Armenia, Azerbaijan, and Georgia.

The distribution of prevalence study sites is shown in Figure 2.

The main data of all retrieved studies reporting on the prevalence of TB in Central Asia and Southern Caucasus are presented in Table 3.

The study designs demonstrate significant variation, ranging from cross-sectional snapshots, such as Alikhanova, N. (2014)’s investigation of 789 TB patients in Azerbaijan, to observational studies like Jenkins, H.E. (2014)’s analysis of 1795 TB cases in Georgia. Sample sizes and geographical coverage also varied. For example, Gadoev, J. (2021) examined 35,122 TB cases in Uzbekistan, while Turbokov studied 95 MDR-TB cases in urban Tashkent. Similarly, Alikhanova’s study in Azerbaijan highlights drug resistance patterns, reporting that, among new and previously treated cases, 42% and 61%, respectively, were resistant to anti-TB drugs, and 13–28% had MDR-TB. Meanwhile, Jenkins HE.’s study in Georgia quantified MDR-TB incidence at 16.2 per 100,000 annually, with distinctions between new and previously treated cases.

Regional studies, such as those focused on Karakalpakstan (Uzbekistan) and Bishkek (Kyrgyzstan), provide granular data that capture localized trends and resistance patterns, which may not align with broader national or international analyses.

DR-TB remains a critical focus in many studies, as highlighted by the differences in the prevalence of MDR-TB and DR-TB. While Jenkins HE. emphasizes MDR-TB trends in Georgia, broader studies, such as Bastard, M. (2018)’s multi-country analysis, underscore the challenges of managing TB in populations with varying levels of resistance.

### 3.3. Prevalence of TB by Sex, Age, and Risk Factors

There were seven studies conducted in Central Asia and in Southern Caucasus countries examining the differences in TB prevalence by gender (Table 4). The authors Bastard M., Sadykova L., and Trubnikov A. identified a significant disparity in prevalence between men and women, with averages of 65.3% and 34.7%, respectively [29,34].

The results on gender distribution are rather controversial. For example, Zanaa A. (2022) shows a near-equal distribution by sex (Male 49.8%, Female 50.2%), which is very similar to the results of Matthias Arnold (2016) and Sadykova L. (2019). However, Bastard M. (2018) shows a significantly higher male proportion among HIV-negative cases, with males representing 68.3% and females 31.7%, suggesting a potential gender bias in certain subpopulations.

Age group categorization differed substantially across studies. For example, Zanaa A. (2022) categorizes early childhood TB cases into narrow age groups, such as 0–1 years (170 cases) and 2–4 years (585 cases), highlighting TB prevalence in very young children. In contrast, Sadykova L. (2019) provides detailed age intervals for adults, including 18–24 years (5770 cases), 25–29 years (5197 cases), and older age groups, which focus more on adult TB prevalence patterns. Studies such as Boldoo T. (2023) and Matthias Arnold (2016), however, only report the mean age of TB cases, 33 years (±17.3) and 30.28 years, respectively, which does not capture age-specific prevalence patterns as effectively. Differences in age and gender distribution complicate the comparison of TB prevalence across studies. For example, Alikhanova N. (2014) reports 71% of new cases among males and 29% among females, with the majority (72%) occurring in individuals aged 15–44 years. In contrast, Jenkins HE (2014) provides prevalence rates per 100,000 population, showing higher rates in males (28.1) than in females (5.5) and significant age-specific differences—17.1 per 100,000 population in the 15–24 age group compared with 4.5 per 100,000 population in those aged 65 and above.

The results from the reviewed articles indicate that the prevalence of TB is higher in patients over 30 years of age (Table 5). Additionally, studies focusing on the pediatric population reported similar prevalence rates, with an average of *n* = 1096 cases among children under 14 years [28,32]. Sadykova’s study reported 8582 cases among young adults aged 20–39 years and 6755 cases among middle-aged individuals aged 40–49 years. Boldoo T. reported a mean age of 40 years (±13.9) among relapse cases.

High-risk groups for TB identified in the studies included individuals deprived of liberty and those with HIV infection. In Bastard M. (2018), 418 cases (30.5%) were HIV-positive. In Sadykova L. (2019), this figure was 490 cases, and in Trubnikov A. (2021) it was 88.4%.

Additionally, Zanaa A. and Gadoev J. investigated the difference in prevalence among urban and rural residents. Their findings revealed that, in Mongolia, the prevalence of TB was higher in urban areas (53.3 per 100,000) [28], while, in Uzbekistan, a greater number of TB cases were reported in rural areas (19,774 cases, accounting for 56% of the total patient population) [32]. However, direct comparisons between these countries are not feasible, as the data do not represent the entire population of each country.

### 3.4. Economics of TB

An overview of the only three studies reporting the economic impact of TB is presented in Table 6. Notably, only one study focused on a specific country (Kazakhstan), while the other two were regional studies. These studies generally addressed different aspects and segments of costs, ranging from monthly to episodic expenditures. The cost of treating MDR-TB was significantly higher than that for treating standard TB. Furthermore, we could not find any data on the costs associated with TB treatment in Turkmenistan, Mongolia, and the Southern Caucasus countries.

## 4. Discussion

We conducted a systematic literature review to identify published data on the prevalence of TB, its associated risk factors, and economic challenges in Central Asia, Southern Caucasus, and Mongolia. In line with the WHO End TB Strategy and the Sustainable Development Goals (SDGs), improving early diagnosis is essential to reduce transmission, initiate timely treatment, and ultimately reduce the TB burden.

Our analysis identified nine studies on the prevalence of TB in this geographic region. Most of the studies (74%) provided data on former Soviet republics such as Kyrgyzstan, Kazakhstan, Uzbekistan, and Tajikistan; however, we did not find data on Turkmenistan, likely because most statistics are considered state secrets and no published data are available [37]. Diagnostic capacity varies across the region. In most Central Asian countries, TB diagnosis is still based on clinical data or simple technologies such as chest X-rays and sputum smear tests [38]. Sputum microscopy is still one of the main methods used to detect MDR in developing countries. Despite its low sensitivity (45–65%) compared with molecular tests, sputum smear microscopy remains the most widespread method [39]. GeneXpert MTB/RIF has improved detection of drug resistance, but access is limited, with only 58% of facilities equipped, mostly in urban regions.

The diagnostic limitations described above are reflected in the heterogeneous epidemiological areas [40]. Traditional culture methods delay diagnosis by 3–8 weeks, while GeneXpert provides results within two hours [41]. These delays hinder timely treatment and increase transmission patterns observed across the region. Our results indicate that the prevalence of TB in Central Asia, Southern Caucasus, and Mongolia remains high, as reported by the WHO [23]. However, rates vary significantly across countries and territories. For instance, Sakko et al. (2019) report that the incidence of TB in Kazakhstan decreased from 227 to 15.2 cases per 100,000 people between 2014 and 2019, while all-cause mortality among TB patients rose from 8.4 to 15.2 per 100,000 people during the same period [42]. TB incidence in Mongolia showed an increasing trend from 41.65 to 55.63 per 100,000 population between 2016 and 2018 [43]. In our review, TB incidence rates in Mongolia declined from 2015 to 2018, but increased again in 2019 to 133 per 100,000 people [31]. The prevalence of MDR-TB ranged from 19% to 26% among newly treated TB patients and from 60% to 70% among previously treated TB patients. Based on these results, it can be concluded that there is a need for evidence-based research to identify and outline vulnerable population groups.

Analysis of notification rates from 2000 to 2017 revealed diverse trends in drug-resistant TB among these countries. While Armenia and Georgia have shown stable notification rates for drug-resistant TB among new cases in recent years, Azerbaijan experienced an increase [11]. In 2017, Armenia had 812 cases (27.1 per 100,000 population), Azerbaijan had 7129 (67 per 100,000 population) and Georgia had 2927 notified cases (69 per 100,000 population). The rate in most EU countries is under 10 per 100,000 population [44]. Countries in Central Asia and the Southern Caucasus inherited a centralized TB diagnostic approach from the Soviet healthcare system, heavily relying on fluorographic screening and networks of specialized TB dispensaries [45]. Diagnostic performance varies by method and setting. Smear microscopy demonstrates variable sensitivity, typically ranging from 47% to 60%, and is less reliable in smear-negative or HIV-co-infected cases. In contrast, Xpert MTB/RIF shows consistently higher sensitivity (89%) and specificity (99%) for pulmonary TB detection, with 95% sensitivity and 98% specificity for rifampicin resistance, as reported by the WHO. Similar findings were observed in a multicenter study in Indonesia, where Xpert achieved 97.4% sensitivity, confirming its effectiveness in real-world settings [23,46].

In Kyrgyzstan, the authors documented that 78.2% of suspected cases were diagnosed with TB through a microscopic examination of a sputum smear [47]. They also confirmed that Xpert MTB/RIF had significantly higher sensitivity and specificity for the detection of TB than microscopic examination of sputum smears in the Kyrgyz population [47].

In Kazakhstan, the implementation of GeneXpert MTB/RIF significantly reduced diagnostic delays, with the time to testing from initiation of MDR treatment reduced to approximately one week. In Uzbekistan, there has been a substantial scale-up of molecular diagnostic capacity, with GeneXpert instruments increasing from six to sixty-seven between 2012 and 2019 and the annual testing capacity expanding from 5574 to 107,330 assays [48].

Factors that were more likely to be associated with TB infection included age > 30 years, male sex, urban residence, and previous TB infection [49].

In Central Asia, TB incidence is higher among middle-aged individuals (ages 20–55), whereas, in other countries, such as Brazil, it is more prevalent among younger populations. The odds of acquiring TB increased significantly with age in young females (ages 5–14 years), while, in young adult males (ages 15–39 years), the increase in odds was marginally significant [50]. A study conducted in Myanmar found that the highest proportion of individuals diagnosed with TB was in the 16–20 year age group [51]. In Georgia, the median age at death for individuals who died following TB treatment was 64.0 years [52].

Sex and age were shown to be important risk factors for TB prevalence in the literature we reviewed. These factors are also often discussed in other studies across the world. For instance, it was shown that gender differences in prevalence among TB patients have been shown to be higher in males than females globally, averaging 9.39 per 100,000 and 7.95 per 100,000, respectively [53]. In most age groups, males had higher disability-adjusted life year (DALY) rates than females, with the burden of MDR-TB increasing with age [53]. Gender may influence the development and progression of TB due to differences in social roles, risk behaviors, and activities. Alcohol and tobacco dependence are more common in men, exacerbating the initial clinical presentation at the time of TB diagnosis [54].

TB has been found to be more prevalent in areas with high population density [53]. Numerous studies indicate that population density is a significant factor influencing the prevalence of childhood TB [28,55,56]. In Uzbekistan, TB is most commonly reported in rural areas among adults [32]. Globally, the prevalence of TB is considerably higher in urban areas than in rural areas [40]. However, in some countries, these statistics are reversed, with individuals residing in rural areas at much higher risk of contracting TB disease. Similarly, in other developing countries where large portions of the population are rurally located, TB incidence in rural areas is greater than or equal to that in large urban locales. Previous studies in South Africa revealed that rural populations are significantly more likely to experience higher levels of transmission compared with their urban counterparts. Urban residents often live closer to healthcare facilities and have easier access to TB treatment and care, which can reduce the duration of the disease. Additionally, rural populations may have fewer venues for social gatherings (e.g., churches, alcohol-related establishments), leading to a higher concentration of individuals in limited locations, which may contribute to increased transmission compared with urban settings [57].

The prevalence of several co-existing illnesses or risk factors was higher in the TB cohort than in the overall Georgia population in 2014 (our study midpoint), including HIV (10% vs. 0.5%), homelessness (10% vs. 0.12%), excess alcohol use (15% vs. 5.3%), and diabetes mellitus (12% vs. 11%) [52].

Ulmasova et al. (2013) reported that, in Uzbekistan, only 34% of MDR-TB cases were detected due to limited laboratory capacity and diagnostic delays [58].

Although the Xpert MTB/RIF test has high sensitivity and specificity for the detection of TB cases, there are some difficulties in scaling up the test nationwide in Kyrgyzstan. The GeneXpert device requires a stable electricity supply, a temperature below 30 °C, and yearly calibration, which is recommended by the manufacturer. Cartridges also need to be stored at temperatures between 2 and 28 °C. Under some conditions, generator instalment is necessary for the GeneXpert device to maintain a continuous electricity supply. However, these conditions cannot always be met in rural areas of Kyrgyzstan [47].

Household illness costs were identified as the main barrier to accessing tuberculosis diagnosis and treatment in Tajikistan [59], where patients face average costs of about USD 4900 purchasing power parity per TB episode in a country with a per capita gross domestic product of USD 1300 [59]. Studies in Kazakhstan also confirm that patients with multidrug-resistant TB face catastrophic costs [36], while research in Mongolia revealed catastrophic household expenditures for TB treatment [60]. Costs in the first 2 months of treatment, including the period of diagnosis and treatment initiation, constitute the most powerful barrier for people with TB to completing the diagnostic process [61].

In Central Asia, the Southern Caucasus, and Mongolia, TB stigmatization creates serious barriers to timely medical care-seeking, as “TB has a large social impact on individuals as high levels of stigma” [62] prevent patients from consulting doctors due to fear of social isolation. Tajikistan has the highest TB incidence in Central Asia [63], while underfunding of the healthcare system and accessibility issues exacerbate delays in diagnosis and treatment. Given the unique characteristics of the former Soviet Union—one of the world’s largest geographical regions with low population density—public health challenges in this region differ from those elsewhere. This context makes the introduction of biomedical technologies related to digitalization, biosensors, and other innovations potentially more cost-effective. Importantly, the process of digitizing the healthcare system must continue, as many post-Soviet countries still lack comprehensive health information systems [64].

There is a direct correlation between delays in TB diagnosis and adverse disease outcomes. Studies show that 5.8% of patients with long diagnostic delays had adverse outcomes compared with 5.1% of patients with short delays [65], demonstrating a statistically significant relationship between diagnostic timeliness and prognosis.

In Mongolia, diagnostic delays are particularly critical. The median total delay time is 37 days (interquartile range 19–76 days) [66] from symptom onset to treatment initiation. Despite positive trends—TB mortality in Mongolia decreased from 4.9 per 100,000 population in 1996 to 2.0 in 2022 [67]—diagnostic delays continue to affect disease control effectiveness.

Under-diagnosis creates specific challenges. In some high-TB-burden countries, 40% of people with TB remain undiagnosed [68], leading to distortion of true statistical indicators of prevalence and mortality. This means that official statistics may substantially underestimate the real disease burden in Central Asia, the Southern Caucasus, and Mongolia, where detection systems face additional barriers, including stigmatization and limited access to healthcare.

Late diagnosis not only increases the risk of adverse outcomes for individual patients but also contributes to further community transmission, creating a vicious cycle of epidemiological underestimation and inadequate public health planning. This delayed recognition undermines both individual patient care and broader disease control efforts in these regions.

TB diagnostic services were severely disrupted by the COVID-19 pandemic, with global testing volumes declining by 21% in 2020 [23]. In Kazakhstan and Kyrgyzstan, GeneXpert systems were repurposed for COVID-19, reducing the TB testing capacity by 35–40%**,** while, in Mongolia, the average diagnostic delay increased from 21 to 65 days, potentially exacerbating transmission.

Diagnostic services were particularly affected, with testing volumes dropping by 21% in 2020 compared with 2019 [23]. In Mongolia, national surveillance data from 2018 to 2021 indicate a median total delay of 37 days from symptom onset to treatment initiation, with the majority attributed to patient delays (median 23 days) rather than health-system delay [31].

During the COVID-19 pandemic, globally there was a significant decline in the detection of TB cases. According to experts, the decrease in the number of new cases is not due to a decrease in the incidence of the disease, but rather to incomplete detection of cases. As a result, there is every reason to believe that the number of cases may increase in the coming years [69]. As the coronavirus pandemic impacted people’s lifestyles in 2020 (Appendix A), the number of confirmed TB cases dropped significantly before rising rapidly again in 2021 [70].

The COVID-19 pandemic led to a catastrophic reduction in TB detection and diagnosis worldwide. According to preliminary WHO data from 84 countries, 1.4 million fewer people received TB care in 2020 compared with 2019—a 21% decline [71]. The number of people newly diagnosed with TB fell from 7.1 million in 2019 to 5.8 million in 2020 and 6.4 million in 2021 [72].

In the WHO European Region, which includes Central Asia and the Southern Caucasus, the situation was particularly severe. TB case notifications declined by 35.5% in the second quarter of 2020 compared with the same period in 2019, which is six times higher than the average annual decline of 5.1% recorded in 2015–2019 [68]. Additionally, there was a 15% decline in the number of people receiving MDR/RR-TB treatment between 2019 and 2020 [73].

Our systematic review revealed limited published data on the impact of COVID-19 on TB in Central Asia. No research on this topic was found in Mongolia, Turkmenistan, or the Southern Caucasus. However, a study by Gabdullina M. and colleagues in Kazakhstan found a significant change in TB indicators during the pandemic. Incidence increased from 31% to 39%, TB among people aged 60+ rose from 16% to 22%, adverse outcomes increased from 11% to 20%, and mortality increased from 6% to 9% [74]. In Tajikistan, TB case registration decreased by 28% in 2020 compared with 2019, reducing treatment coverage [75].

Globally, there was a 3% increase in DR-TB cases between 2020 and 2021. There were 450,000 new cases of RR-TB in 2021 [76].

Sachin Silva and co-authors examined the economic impact of TB mortality in 120 countries. From 2020 to 2050, with the current 2% annual decline in TB mortality, there would be an estimated 31.8 million TB deaths (95% uncertainty interval: 25.2 to 39.5 million), resulting in economic losses of USD 17.5 trillion (USD 14.9 to 20.4 trillion). However, if the SDG target for TB mortality is met by 2030, an estimated 23.8 million TB deaths (USD 18.9 to 29.5 million) and economic losses of USD 13.1 trillion (USD 11.2 to 15.3 trillion) could be avoided [77].

However, we could not find sufficient evidence on the economic impact of TB prevalence in the countries of interest. There is a lack of in-depth data on the costs associated with TB treatment, including direct medical expenses, indirect costs, and the broader socioeconomic consequences. Addressing these gaps through further research is crucial for developing effective policies and resource allocation strategies.

## 5. Conclusions

In conclusion, TB incidence in Central Asia, Southern Caucasus, and Mongolia varies considerably across countries, with some regions experiencing significantly higher rates than others. The burden of TB is more pronounced among men than women, and the disease predominantly affects individuals of working age, with fewer cases reported in children and a moderate prevalence among the elderly.

Despite a decrease in the absolute number of cases, TB prevalence in the region remains exceptionally high compared with Europe and America, continuing to pose a major concern. This situation necessitates substantial financial expenditures from both governments and patients, largely due to the socioeconomic challenges of the population.

Key risk factors for TB transmission include HIV-positive status, comorbidities such as diabetes, and undernutrition. Social determinants like unemployment, imprisonment, and close contact with TB patients also significantly contribute to the spread of the disease.

The economic burden of TB, particularly of MDR-TB, is substantial, with treatment costs straining healthcare systems across the region. The financial impact varies by country, with higher costs associated with managing drug-resistant strains of TB. These trends underscore the urgent need for more effective prevention, early detection, and management strategies to reduce both the health and economic impacts of TB in Central Asia, the Southern Caucasus, and Mongolia.

## 6. Strengths and Limitations

The strength of this study is that it is the first systematic review of TB prevalence in Central Asia, Southern Caucasus, and Mongolia.

An extensive literature review was conducted by two independent reviewers in accordance with PRISMA guidelines. Furthermore, the protocol for this study was registered in advance with PROSPERO, and we included articles from peer-reviewed journals in both English and Russian. The Newcastle–Ottawa Scale was utilized for quality assessment.

Our study identified several limitations. The lack of data on the prevalence of TB in Turkmenistan significantly reduces the objectivity and representativeness of the analysis results. Additionally, the diversity of methodologies and study designs employed across different countries poses challenges for generalizing the findings. Another limitation of our review is that a considerable number of potentially relevant articles could not be included due to the unavailability of full texts. This may have restricted the comprehensiveness of the evidence synthesis. Despite the broad scope of the literature search, the total number of available studies may be insufficient to draw comprehensive conclusions for the region as a whole.

## 7. Recommendations for Future Research and Policy

Despite a decrease in cases, TB prevalence in the studied countries remains high compared with Europe and America, requiring significant financial resources due to the population’s socioeconomic challenges. Further research is recommended to expand studies on TB in Central Asia, Southern Caucasus, and Mongolia, particularly by including Turkmenistan, to gain a more comprehensive understanding of TB dynamics in the region. Additionally, more in-depth investigations are needed to assess the impact of factors such as poverty, healthcare access, and social conditions on TB prevalence.

To enhance TB control efforts, it is crucial to establish reliable registers that provide accurate data on TB prevalence, improve data comparability between countries, and better identify vulnerable groups. These measures will contribute to the successful recognition and treatment of TB. Furthermore, ongoing research is essential to examine the long-term effects of the COVID-19 pandemic on TB diagnosis, treatment, and outcomes, ensuring the development of effective interventions to reduce TB mortality in the post-pandemic period.

## Figures and Tables

**Figure 1 diagnostics-15-02314-f001:**
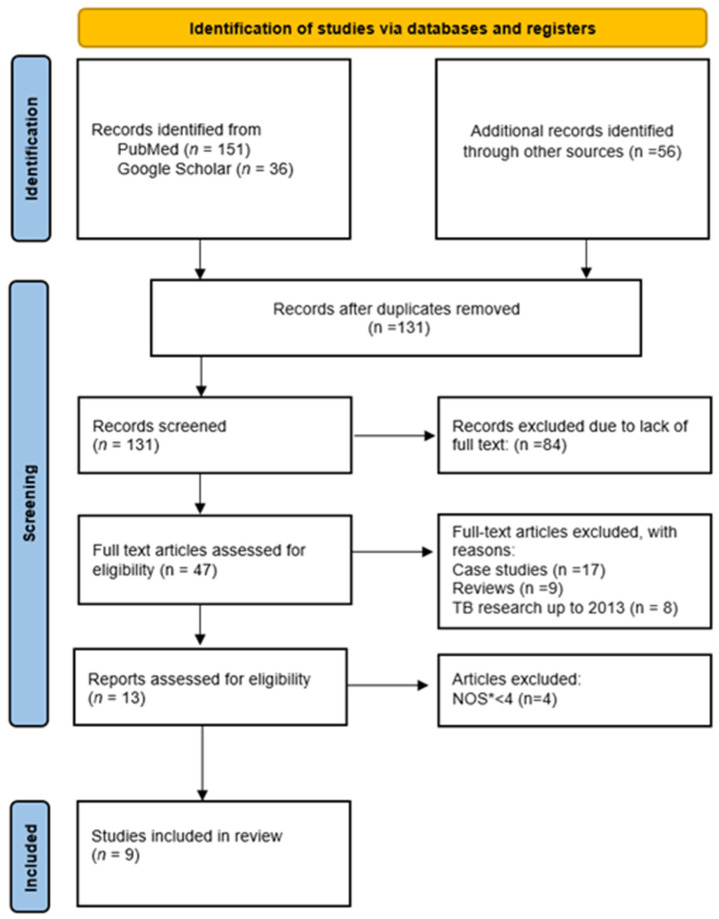
Flowchart of the literature search strategy. * NOS, Newcastle–Ottawa Scale.

**Figure 2 diagnostics-15-02314-f002:**
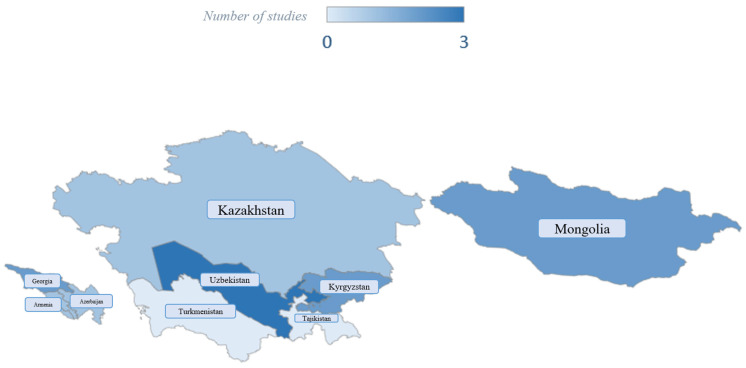
Number of studies on tuberculosis prevalence in Central Asia and Southern Caucasus. (1) Uzbekistan (3) [29,30,31,32]; (2) Mongolia, Kyrgyzstan, and Georgia (2) [33]; (3) Mongolia (among children) [28]; (4) Armenia, Kazakhstan, and Azerbaijan (1) [34]; (5) Tajikistan and Turkmenistan (0).

**Table 1 diagnostics-15-02314-t001:** Quality assessment according to the Newcastle–Ottawa scale (NOS), score ≥ 4, *n* = 9.

Article (First Author, Year)	Selection (Max 5 *)	Comparability (Max: 2 *)	Outcome (Max. 3 *)	Total (Max: 10 *)
Zanaa A, 2022	***	**	***	8 *
Alikhanova N, 2014	***	**	***	8 *
Gadoev J, 2021	***	*	**	6 *
Bastard M, 2018	**	*	*	6 *
Matthias Arnold, 2016	**	**	*	5 *
Sadykova L., 2019	**	*	**	5 *
Jenkins HE, 2014	**	**	*	5 *
Boldoo T, 2023	**	*	*	4 *
Trubnikov A., 2021	**	*	*	4 *

Note: In the Newcastle–Ottawa Scale (NOS), each asterisk (*) corresponds to one point awarded for the corresponding criterion. Accordingly: *** = 3 points, ** = 2 points, * = 1 point. The maximum scores are: Selection = 5, Comparability = 2, Outcome = 3, Total = 10.

**Table 2 diagnostics-15-02314-t002:** Inclusion and exclusion criteria for study selection.

Inclusion Criteria	Exclusion Criteria
1. Original articles describing TB indicators 2. Articles published before April 2023. 3. Case-control studies, randomized control trials, prospective and retrospective cohort studies, and series were eligible for inclusion.	1. Studies related to extrapulmonary TB cases 2. Studies describing a specialized population (prisoners) 3. Abstracts, published articles not related to the review topic, and systematic reviews/meta-analyses 4. TB research up to 2013 5. Case reports and case studies

**Table 3 diagnostics-15-02314-t003:** Studies providing data on the prevalence of TB in Central Asia and Southern Caucasus (*n* = 9).

Article (First Author, Year)	Study Design	Sample Size	Country /Region	Type of TB	TB Prevalence/Incidence
Zanaa, A., 2022 [28]	Retrospective descriptive study	4242 cases of TB	Mongolia	(1) clinically active TB, (2) not clinically active TB, (3) bacteriologically confirmed TB, (4) presumptive TB, (5) no TB exposure	TB was higher among children aged 5–14 years (68.5% of total childhood TB cases between 2014 and 2020), the number of cases of multidrug-resistant TB was 14.1 (±5.0). MDR-TB was 3.1% of all childhood TB cases
Boldoo, T., 2023 [31]	Retrospective descriptive study	-	Mongolia	Bacteriologically confirmed TB, extrapulmonary TB, clinically diagnosed TB, and other previously treated TB.	TB cases were registered as 133 per 100,000, MDR-TB cases 211 per 100,000, XDR-TB cases 7 per 100,000
Matthias Arnold, 5 April 2016 [30]	Cross-sectional data	139 patients with TB	Kyrgyzstan, Bishkek	TB general	In 2013, TB prevalence was 190 per 100,000
Gadoev, J., December 2021 [32]	16-year cohort study	All patients with TB—35,122	Republic of Karakalpakstan, Uzbekistan	TB (pulmonary TB) 29,130 (83) EPTB (extrapulmonary TB) 5992 (17) TB treatment category I (2(3)HRZE(S)/4 H3R3) a 27,465 (79) category II (2HRZES/1(2)HRZE/5 H3R3E3) b 7497 (21) category III (2HRZ/4 H3R3) c 160 (<1)	69 cases per 100,000 population
Bastard, M., 8 March 2018 [34]	Retrospective cohort study	1369	7 countries: Abkhazia, Armenia, Colombia, Kenya, Kyrgyzstan, Swaziland, and Uzbekistan.	DR-TB	MDR-TB among new TB cases ranged from 11.0% in Armenia to 32.0% in Kyrgyzstan, and, among previously treated TB cases, ranged from 33.0% in Georgia to 63.0% in Uzbekistan.
Sadykova, L., June 2019 [29]	Retrospective cohort study with continuous sampling.	36,926 TB cases	Kazakhstan	_	67 cases per 100,000 population, and the incidence of MDR-TB or rifampicin-resistant TB was 39 cases per 100,000 population
Trubnikov, A., 13 April 2021 [33]	A cohort study involving secondary analysis of routinely collected data.	95 patients with laboratory-confirmed MDR-TB.	Uzbekistan	RR/MDR TB	Based on routine surveillance data in 2018, out of 2238 registered MDR-TB patients, 137 (6.1%) were enrolled in short treatment regimens (STRs). In 2019, out of 2060 MDR-TB cases, 157 (7.6%) received STR treatment
Alikhanova, N., 21 October 2014 [27]	A cross-sectional study	789 patients (549 new and 240 previously treated)	the Republic of Azerbaijan	DR-TB	Among all new and previously treated cases, respectively 231 (42%) and 146 (61%) were resistant to any anti-tuberculosis drug, and 72 (13%) and 66 (28%) had MDR-TB.
Jenkins, H.E., 20 March 2014 [35]	An observational epidemiological study	1795	Georgia	MDR TB	Average annual MDR-TB notified incidence was 16.2 per 100,000 (3.2 new MDR-TB cases per 100,000 and 12.8 previously treated MDR-TB cases per 100,000).

**Table 4 diagnostics-15-02314-t004:** Studies providing data on the prevalence of TB by sex and age, *n* = 9.

Article (First Author, Year)	Male (%)	Female (%)	Age Group (%)
Zanaa A, 2022 [28]	*n* = 2111 (49.8%)	*n* = 2131 (50,2%)	(1) 0–1 years (*n* = 170; 4) 2–7 years (*n* = 586; 13.8%), 8–14 years (*n* = 668; 15.7%). (2) 0–4 years (*n* = 886; 31.4%). 5–14 years (*n* = 1932; 68.6%);
Boldoo T, 2023 [31]	New cases 57.2%—a mean age of 33 (±17.3) years. relapse cases a mean age of 40 (±13.9) years.	New RR/MDR-TB cases, 40.4%, similar to the proportion seen in all TB notifications.	A mean (±standard deviation (SD)) age of 33 (±17.3) years, whereas 66.9% of relapse cases were male, with a mean age of 40 (±13.9) years. In 2019, 9.1% (*n* = 415) of TB notifications were aged under 15 years and 2.7% (*n* = 121) were aged under 5 years
Matthias Arnold, [30] 5 April 2016	Male—49.64%	Female—50.36%	A mean age of 30.28 (28.14:32.43)
Gadoev J, 5 December 2021 [32]	18,032 (51%)	17,090 (49%)	Children (0–14)—2339 (7%) Teenagers (15–18 years)—2038 (6%) Adults (19–55 years)—24,394 (69%) Seniors (over 55 years)—6351 (18%)
Bastard M, 8 March 2018 [34]	HIV-negative total *n* = 951 *n*(%)—683 (71.8) HIV-positive *n* = 418 *n*(%)—195 (46.6) Total *n* = 1369 *n*(%)—878 (64.1)	HIV-negative total *n* = 951—*n*(%)—268 (28.2) HIV-positive total *n* = 418—*n*(%)—223 (54.4) Total *n* = 1369 *n*(%)—491 (35.9)	<35 *n* = 681 (50.1), HIV(+) = 236 (57.3) HIV(-) = 445 (47.0) ≥35 *n* = 677 (49.9), HIV(+) = 176 (42.7) HIV(-) = 501 (53.0)
Sadykova L. June 2019 [29]	*n* = 22,648 (61.3)	*n* = 14,278 (38.7)	18–24 *n* = 5770 (15.6) 25–29 *n* = 5197 (14.1) 30–39 *n* = 8582 (23.2) 40–49 *n* = 6725 (18.2) 50–59 *n* = 5613 (15.2) 60 < *n* = 5039 (13.6)
Trubnikov A. 13 April 2021 [33]	Total male *n* = 67 (70.5%), new cases—46 (48.4%)	Total female *n* = 28 (29.5)	<40 *n* = 33 (34.7) ≥40 *n* = 62 (65.3)
Alikhanova N, 21 October 2014 [27]	Total male *n* = 338 (71%),	Total female *n* = 138(29%)	15–24 *n* = 122 (26%) 25–34 *n* = 117 (25%) 35–44 *n* = 102 (21%) 45–54 *n* = 68 (14%) 55–64 *n* = 45 (9%) 65≤ *n* = 22 (5%)
Jenkins HE, 20 March 2014 [35]	Male 28.1 per 100,000 population	Female 5.5 per 100,000 population	0–4—1.7 per 100,000 5–9—1.5 per 100,000 15–24—17.1 per 100,000 25–34—33.9 per 100,000 35–44—27.8 per 100,000 45–54—18 per 100,000 55–64—12 per 100,000 65≤—4.5 per 100,000

**Table 5 diagnostics-15-02314-t005:** Studies providing data on risk factors for tuberculosis, *n* = 7.

Article (First Author, Year)	Risk Groups/Factors	
Zanaa A, 2022 [28]	Difference in the city and village Central region—2020 53.3 per 100,000 children Other regions—2020 11.7 per 100,000 children	
Matthias Arnold, [30] 5 April 2016	Socioeconomic factors as the main causes of TB infections	
	Education (%) No schooling—0.72 Primary—2.88 Secondary—64.75 Tertiary—30.22	Employment (%) Unemployed—24.46 Informal—35.97 Formal—23.74 Retired—5.76
Gadoev J, 5 December 2021 [32]	Place of residence Urban—9289 (26) Rural—19,774 (56) Missing data—6059 (17) Social characteristics Worker—1605 (5) Employee—1440 (4) Student—2572 (7) Disabled—819 (2) Retiree—3577 (10) Unemployed—15,409 (44) Missing data—8904 (25)	HIV status HIV positive—19 (<1) HIV negative—26,373 (75) Missing data—8730 (24) Former prisoner No—2582 (7) Yes—9 (<1) Missing data—32,531 (92) Contact with a patient with tuberculosis Yes—23,417 (67) No—1915 (5) Missing data—9790 (28)
Bastard M., 8 March 2018 [34]	HIV status Negative: *n* = 951 Positive: *n* = 418 Former prisoner No—1214 (88.7) Yes—155 (11.3) Contact with MDR-TB case No—1241 (90.7) Yes—127 (9.3)	None—1 Diabetes—70 (5.7) BMI <18.5—347 (34.4) ≥18.5–663 (65.6) None—359
Sadykova L. June 2019 [29]	1. Drug dependence *n* = 104 (0.28) 2. Contact with TB patients *n* = 559 (1.51) 3. Diabetes *n* = 988 (2.68) 4. Alcoholism *n* = 1742 (4.72) 5. Pregnancy and postpartum period *n* = 964 (2.61)	6. Being in prison in the last 2 years *n* = 383 (1.03) 7. HIV *n* = 490 (1.32) 8. Two risk factors *n* = 204 (0.55) 9. More than 2 risk factors *n* = 24 (0.06) 10. No data *n* = 31,468 (85.22)
Trubnikov A. 13 April 2021 [33]	HIV status HIV positive—84 (88.4%) HIV negative—11 (11.6%) BMI <18.5–58 (74.4) ≥18.5–21 (26.9) Absent—16	Any concomitant pathology—56 (58.9) Diabetes—18 (28.1) Hepatitis—19 (29.7) Anemia—33 (51.6)
AlikhanovaN, 21 October 2014 [27]	Social status Unemployed—374 (79) Working—41 (9) Retired—40 (8) Disabled—8 (2) Student—13 (3) Living conditions Owner—435 (91) Renting—29 (6) Hostel—5 (1) Homeless—7 (2) Financial status Low income—248 (52) Middle income—218 (46) High income—2 Unknown—8 (1)	Smoking status No—243 (51) Yes—233 (49) Alcohol use No—333 (70) Yes—143 (30) Drug use No—471 (99) Yes—0 Unknown—5 (1) Prison history * No—465 (98) Yes—10 (2) Unknown—1 HIV status Negative—397 (84) Positive—1 Unknown—78 (16)

* History of incarceration at any time in the past.

**Table 6 diagnostics-15-02314-t006:** Studies providing data on the cost of tuberculosis treatment in Central Asia, *n* = 3.

Article (First Author, Year)	Study Design	Sample Size	Country /Region	Characteristics of the Population (Age, Gender)	Type of TB	Economic Impact Data
Matthias Arnold, [30] 5 April 2016	Cross-sectional data	139	Kyrgyzstan, Bishkek	30 years (*p* < 0.001). Female—50.36 Male—49.64	TB general	Equivalence income is highest in TB patients with KGS 4704 (USD 106)
Susan van den Hof, 2016 [36]	Structured interview	148	Kazakhstan	A total of 54 patients with TB and 94 patients with MDR-TB in Kazakhstan	TB general; MDR-TB	For diagnostics and the current episode of treatment of patients with TB, the cost was USD 929; for patients with MDR-TB, it was USD 3125
Stefan Kohler, 2021 [22]	-	-	Uzbekistan, Karakalpakstan	Treatment plans (4-, 6-, 9-, and 20-month programs)	(1) DS-TB (2) MDR-TB	A 6-month course of treatment for TB and a 20-month course of treatment for MDR-TB was USD 1401 ± 274 thousand per year, to which was added the import of drugs in the amount of USD 34 ± 6.4 thousand per year.

## Data Availability

The datasets used and analyzed in the current study are available from the corresponding author on reasonable request.

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
