# Peer review of "Prevalence of Tuberculosis in Central Asia and Southern Caucasus: A Systematic Literature Review [Author-notes fn1-diagnostics-15-02314]"

_diagnostics, 2025, doi:10.3390/diagnostics15182314_

Round 1
Reviewer 1 Report
Comments and Suggestions for Authors
I would like to thank you for the opportunity to review such an interesting and relevant paper. The article is very well structured and raises important data for public health assessment. Below, I present small considerations and questions.
Line 44: Include the meaning of the acronym WHO and include only the acronym on line 49. Update the WHO reference to the most current 2024.
Standardize WHO references 5 and 7. Review the other references and standardize according to the journal's instructions.
Line 60: Multidrug-resistant tuberculosis (MDR-TB) The correct acronym is (MDR-TB). Use the acronym for tuberculosis on line 64 and elsewhere in the text.
Lines 74–77: The data used are old, from 2017, even before the COVID-19 pandemic. The reference description is also invalid; it was not possible to identify the data referenced. (Ref. 12)
Line 67: "130 per 100,000 population" - Lines 75: "27.1 per 100,000 people." Standardize the incidences, e.g., "X cases per population."
After the first use of the acronym, maintain this standard throughout the text. Line 190: Enter the meaning of SD.
Line 203: Format the word "Figure 1.", capitalizing the "F." Review the entire text for acronyms and their meanings.
The information in the article is interesting, but the entire text needs to be refined, including the English, especially the introduction.
The results should present only the data obtained; any other comments/analysis regarding the data should be included in the discussion.
Comments on the Quality of English Language
The English text contains disjointed and poorly written sentences, non-academic terms, and informational information that is not standardized. The English needs to be revised, especially in the introduction. Did the authors present a certificate of translation from English?
Author Response
Dear Editor and Reviewers,
Thank you very much for taking the time to review our manuscript and providing valuable feedback. We have carefully considered each point you raised, as they have significantly contributed to the enhancement of our manuscript and the study.
We have inserted line numbers and highlighted the changes made in the manuscript in yellow to make them easily identifiable.
We remain open to any further suggestions you may have and eagerly await your evaluation.
Sincerely yours,
Malika Idayat

Reviewer 2 Report
Comments and Suggestions for Authors
The manuscript entitled "Prevalence of tuberculosis in Central Asia and Southern Caucasus: a systematic literature review" represent a review of recent studies on tuberculosis in Central Asia and Southern Caucasus, where this important disease still has high mortality. The manuscript is prepared well generally but some improvements are required as listed below.
- The "+" sign should be removed in the Title.
- 84 articles were excluded due to the lack of full-text. However, what was the reason that these articles lacked full-text? Or the authors could not reach them? Did they try different ways to obtain these articles? Because the number of articles included is low.
- Some tables contain expressions with Cyrillic alphabet.
- Line 346: "Our analysis revealed that 11 studies"... Should it be 9?
Author Response

(The authors gave the same response as above.)

Round 2
Reviewer 1 Report
Comments and Suggestions for Authors
Dear all, thank you for agreeing to the suggested changes and reviewing the manuscript. The article is clearer, and I hope it will be helpful to readers and the government officials of the regions analyzed.